# Experience Augmented Policy Optimization for LLM Reasoning

**Jinda Lu** [1]  **Kexin Huang** [1]  **Junkang Wu** [1]  **Shuo Yang** [2]  **Jinghan Li** [1]  **Chiyu Ma** [3]  **Shaohang Wei** [2]  **Xiang Wang** [1]
**Guoyin Wang** [3]  **Jingren Zhou** [3]

## Abstract

Reinforcement Learning with Verifiable Rewards (RLVR) is a powerful paradigm for improving the reasoning capabilities of large language models (LLMs). However, existing RLVR methods typically rely on on-policy optimization from scratch, resulting in high sampling costs and inefficient utilization of accumulated experience. As model capabilities and policy behaviors evolve during training, recent attempts to reuse experience via fixed reasoning trajectories further suffer from policy mismatch. Motivated by these limitations, we argue that experience in RLVR should not be reused as fixed reasoning trajectories, but instead expressed in a policy-adaptive manner. In this work, we propose **E**xperience-**A**ugmented **P**olicy **O**ptimization (**EAPO**), which leverages a prior RL-optimized policy as an action-level experience prior and selectively injects experience at critical decision points during rollout. To ensure stable and unbiased learning from experience-augmented rollouts, EAPO further incorporates an adapted importance sampling scheme. Experiments on using Qwen-2.5-math 7b and Qwen-3-8B on five different benchmarks demonstrate that EAPO consistently improves reasoning performance over state-of-the-art RLVR methods.

## 1. Introduction

Reinforcement Learning with Verifiable Rewards (RLVR) has demonstrated strong effectiveness in improving the reasoning capabilities of large language models (LLMs) (Shao et al., 2024; Jaech et al., 2024; Comanici et al., 2025). Through extensive rollouts and feedback from verifiable rewards, RLVR encourages models to produce longer and more reliable reasoning processes, thereby *accumulating*

*reasoning experience* that gives rise to advanced behaviors such as chain-of-thought (CoT) reasoning and self-reflection (Guo et al., 2025; Yang et al., 2025) (Figure 1(a)).

Despite these successes, current RLVR strategies (Yu et al., 2025; Wang et al., 2025; Wu et al., 2025; Shao et al., 2024) typically perform on-policy rollouts and policy optimization *from scratch*, requiring massive amounts of sampling to explore successful trajectories within large search spaces. Crucially, this paradigm *fails to exploit the valuable experience* encoded in prior RL-optimized models ($\pi_{RL}$), resulting in substantial inefficient exploration and, more importantly, imposing a fundamental bottleneck on the scalable improvement of reasoning performance.

To address this, recent work (Zhan et al., 2025) treats previously generated, static reasoning trajectories as experience by incorporating these trajectories into current rollouts. However, as reinforcement learning progresses, the capabilities and behavior of the policy model evolve, causing earlier trajectories to become increasingly mismatched with the current policy.

In this paper, we argue that *experience in RLVR should not be reused in the form of fixed reasoning trajectories, but instead represented in a manner that adapts to the evolving policy*. Prior studies (Wang et al., 2025; Cheng et al., 2025; Huang et al., 2026) suggest that successful reasoning outcomes are often determined by a small number of pivotal actions (tokens), rather than entire trajectories. This observation motivates an action-level view of experience, where reinforcement learning shapes which decisions are more likely to lead to successful outcomes. Through RL optimization, such experience is implicitly encoded in an RL-optimized policy $\pi_{RL}$, which captures a structured prior over critical actions. By comparing the behavior of the current policy with this prior, EAPO identifies decision points where the current policy departs from experience, enabling targeted injection of prior experience during training (Figure 1(b)).

To this end, we propose **E**xperience-**A**ugmented **P**olicy **O**ptimization (EAPO), a framework that leverages a prior RL-optimized policy $\pi_{RL}$ as an action-level prior to assist the training of the current policy $\pi_\theta$. EAPO operates through two key components: (1) experience-guided action resam-

---

[1]University of Science and Technology of China [2]Peking University [3]Independent Researcher. Correspondence to: Xiang Wang <xiangwang1223@gmail.com>.

*Proceedings of the 43rd International Conference on Machine Learning*, Seoul, South Korea. PMLR 306, 2026. Copyright 2026 by the author(s).

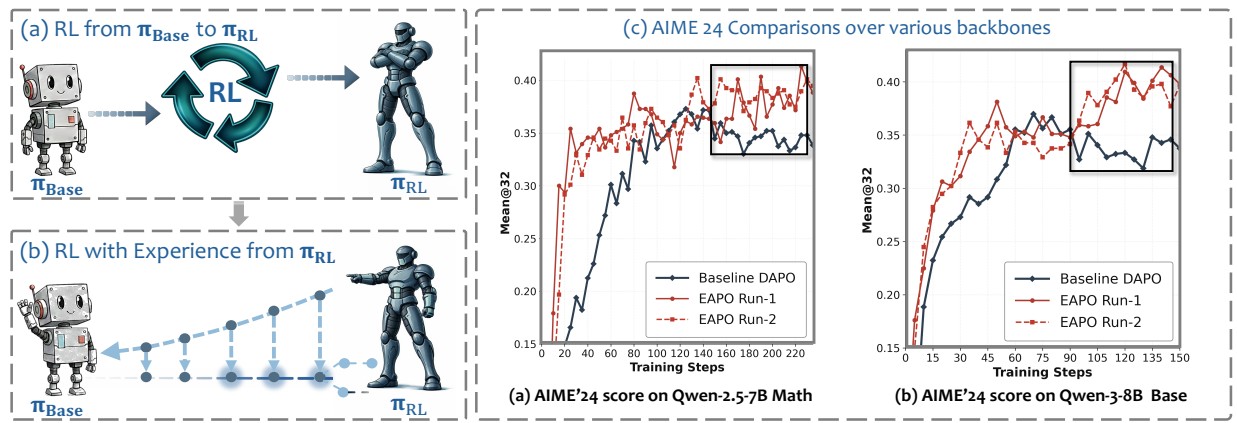

*Figure 1.* **Overview of experience-augmented reinforcement learning.** (a) Standard reinforcement learning with verifiable rewards (RLVR) optimizes a base policy $\pi_{\text{Base}}$ into an RL-optimized policy $\pi_{\text{RL}}$ through iterative rollouts and policy updates. (b) Instead of training from scratch, EAPO leverages experience from a prior RL-optimized policy $\pi_{\text{RL}}$ during rollout of the current policy by resampling actions at critical decision points, introducing experience that adapts to the evolving policy. (c) Empirical comparisons on AIME'24 show that incorporating experience from $\pi_{\text{RL}}$ significantly accelerates exploration and improves accuracy compared to DAPO (Yu et al., 2025), across various base models (Qwen-2.5-7B-Math (Yang et al., 2024b) and Qwen-3-8B (Yang et al., 2025)).

pling during rollout and (2) experience-aware policy optimization for stable learning.

**Experience-guided action resampling.** EAPO identifies critical decision points by comparing the action selection of the current policy $\pi_\theta$ with those of a prior RL-optimized policy $\pi_{\text{RL}}$, and injects experience by resampling actions at these points with $\pi_{\text{RL}}$ during rollout.

**Experience-aware policy optimization.** To ensure stable optimization from experience-augmented rollouts, EAPO employs an experience-aware optimization strategy that adjusts policy updates according to the intensity of experience injection, ensuring robust and effective policy optimization.

As illustrated in Figure 1(c), experiments on AIME24 with Qwen-2.5-7B-Math (Yang et al., 2024b) and Qwen-3-8B (Yang et al., 2025) demonstrate that, compared to DAPO (Yu et al., 2025), EAPO significantly reduces exploration steps and yields stable and consistent improvements across different backbones.

## 2. Preliminaries

In this section, we revisit Reinforcement Learning with Verifiable Rewards (RLVR) and review representative optimization strategies (Shao et al., 2024; Yu et al., 2025).

### 2.1. Reinforcement Learning with Verifiable Rewards

Reinforcement Learning with Verifiable Rewards (RLVR) aims to enhance the reasoning capabilities of large language models (LLMs) by aligning model outputs with automatically verifiable answers. Given a batch of $B$ question–answer pairs $\{(q^b, y^b)\}_{b=1}^{B}$ sampled from dataset $\mathcal{D}$, where

$q^b$ denotes the input question and $y^b$ denotes the corresponding ground-truth answer, the model $\pi_\theta$ generates an output sequence $\boldsymbol{o}^b$ that consists of a reasoning process and a final prediction. The final prediction is enclosed in `\boxed{.}`, while the intermediate reasoning steps are delimited by `<think>...</think>`, which enables automated extraction and verification of the predicted answer against the ground truth. RLVR employs a binary reward function $\mathbf{R}(\cdot)$, which assigns a reward of $1$ if the extracted prediction matches the ground truth and $0$ otherwise.

The objective of RLVR is to maximize the expected reward over the training distribution, which can be formulated as:

$$\mathcal{J}_{\text{RLVR}}(\theta) = \mathbb{E}_{(q^b, y^b) \sim \mathcal{D}} \mathbb{E}_{\boldsymbol{o}^b \sim \pi_\theta(\cdot | q^b)} \left[ \mathbf{R}(\boldsymbol{o}^b, y^b) \right]. \quad (1)$$

### 2.2. RLVR Optimization Algorithms

**Group Relative Policy Optimization (GRPO).** GRPO is a widely adopted optimization strategy for RLVR that stabilizes training by computing advantages relative to a group of responses for the given question (Shao et al., 2024). Specifically, for each input $q^b$, GRPO samples a group of $G$ responses $\{\boldsymbol{o}_i^b\}_{i=1}^{G}$ from the old policy $\pi_{\text{old}}$ and normalizes rewards within the group to estimate advantages. The GRPO objective is defined as:

$$\mathcal{J}_{\text{GRPO}}(\theta) = \mathbb{E}_{(q^b, y^b) \sim \mathcal{D}} \mathbb{E}_{\{\boldsymbol{o}_i^b\}_{i=1}^{G} \sim \pi_{\text{old}}(\cdot | q^b)} \left[ \frac{1}{G} \sum_{i=1}^{G} \frac{1}{|\boldsymbol{o}_i^b|} \sum_{t=1}^{|\boldsymbol{o}_i^b|} \right.$$

$$\min \left( r_{i,t}^b \cdot \hat{A}_{i,t}^b, \text{clip}(r_{i,t}^b, 1 - \epsilon, 1 + \epsilon) \cdot \hat{A}_{i,t}^b \right) \quad (2)$$

$$\left. - \beta \mathbb{D}_{\text{KL}}(\pi_\theta \| \pi_{\text{ref}}) \right],$$

Here, $r_{i,t}^b$ denotes the importance sampling ratio between

the current policy $\pi_\theta$ and the old policy $\pi_{\text{old}}$:

$$r_{i,t}^b = \frac{\pi_\theta(\boldsymbol{o}_{i,t}^b \mid \mathrm{q}^b, \boldsymbol{o}_{i,<t}^b)}{\pi_{\text{old}}(\boldsymbol{o}_{i,t}^b \mid \mathrm{q}^b, \boldsymbol{o}_{i,<t}^b)}. \quad (3)$$

The advantage $\hat{A}_{i,t}^b$ is computed by normalizing rewards within each response group:

$$\begin{cases} \hat{A}_{i,t}^b = \dfrac{\mathbf{R}_i^b - \text{mean}(\mathbf{R}^b)}{\text{std}(\mathbf{R}^b)}, \\ \mathbf{R}_i^b = \mathbb{I}\big(\texttt{is\_equivalent}(\boldsymbol{o}_i^b, y^b)\big), \end{cases} \quad (4)$$

where $\mathbb{I}(\cdot)$ is the indicator function.

**Decoupled Clip and Dynamic Sampling Policy Optimization (DAPO).** DAPO (Yu et al., 2025) improves upon GRPO by removing KL regularization and introducing asymmetric clipping, dynamic sampling, and token-level loss normalization, leading to new state-of-the-art performances. Formally, the DAPO objective is given by:

$$\mathcal{J}_{\text{DAPO}}(\theta) = \mathbb{E}_{(\mathrm{q}^b, y^b) \sim \mathcal{D}} \mathbb{E}_{\{\boldsymbol{o}_i^b\}_{i=1}^G \sim \pi_{\text{old}}(\cdot|\mathrm{q}^b)} \left[ \frac{1}{\sum_{i=1}^G |\boldsymbol{o}_i^b|} \sum_{t=1}^{|\boldsymbol{o}_i^b|} \min \right.$$
$$\left. \left( r_{i,t}^b \cdot \hat{A}_{i,t}^b, \text{clip}\left(r_{i,t}^b, 1 - \epsilon_{\text{low}}, 1 + \epsilon_{\text{high}}\right) \cdot \hat{A}_{i,t}^b \right) \right], \quad (5)$$

Here, $\epsilon_{\text{low}}$ and $\epsilon_{\text{high}}$ decouple the clipping thresholds for negative and positive advantages, respectively, allowing more aggressive updates for high-quality responses.

## 3. Approach

In this section, we present **E**xperience-**A**ugmented **P**olicy **O**ptimization (EAPO), which consists of three core components: **(1) Experience formalization**, where prior RL experience is interpreted as a prior over critical actions; **(2) Experience-guided resampling**, which incorporates such experience during rollout via sparse, token-level intervention; and **(3) Experience-aware optimization**, which integrates experience-augmented trajectories into stable on-policy learning. The algorithm of EAPO is in Alg. 1.

### 3.1. Experience as a Prior over Critical Actions

Experience has been increasingly recognized as a key driver of continual progress in reinforcement learning, enabling models to move beyond repeatedly exploring from scratch and leading to more stable learning (Silver & Sutton, 2025). In the context of large language models' reasoning, work in (Zhan et al., 2025) treats previously generated model responses as experience and reuses them during subsequent rollouts. However, as the policy continuously evolves during reinforcement learning, reasoning trajectories generated at earlier stages often become mismatched with the current policy, rendering such static responses unreliable sources.

Based on this observation, we argue that experience in LLM reasoning should be characterized at the *action level*, in terms of emphasis over critical decisions. Through extensive on-policy optimization, an RL-optimized model $\pi_{\text{RL}}$ does not merely memorize specific reasoning paths, but instead learns which actions are more likely to lead to successful outcomes under particular decision contexts. Such experience is implicitly encoded in the parameterized distribution of $\pi_{\text{RL}}$, endowing it with strong generalization capability.

Accordingly, in EAPO, we treat the prior RL-optimized policy $\pi_{\text{RL}}$ as a structured, adaptable representation of experience. By comparing the action distributions of the current policy $\pi_\theta$ and $\pi_{\text{RL}}$, we identify critical decision points where the current policy may deviate from effective decisions, and incorporate prior RL experience at these points with minimal intervention. In this way, EAPO elevates experience from static samples to a *generalizable decision prior*, while preserving the on-policy nature of policy optimization.

### 3.2. Experience-guided Resampling

EAPO injects experience only when the current policy is overconfident yet disagrees with $\pi_{\text{RL}}$, enabling sparse and targeted correction during rollout.

**Critical token identification.** At each decoding step $t$, given the input question $q$ and the current generated prefix $y_{<t}$, we compare the token-level behavior of the current policy $\pi_\theta$ with that of a previously optimized RL policy $\pi_{\text{RL}}$. Specifically, let $y_t$ denote the token sampled by $\pi_\theta$ at step $t$. We define the token-level discrepancy $\delta_t$ as the log-likelihood ratio between current and prior policies:

$$\delta_t = \log \pi_\theta(y_t \mid q, y_{<t}) - \log \pi_{\text{RL}}(y_t \mid q, y_{<t}). \quad (6)$$

This discrepancy characterizes the degree of disagreement between the current on-policy and prior RL experience.

When $\delta_t \approx 0$, the two policies assign similar probabilities to the selected token, indicating that the current decision is consistent with RL experience. Similarly, when $\delta_t < 0$, the RL-optimized policy assigns a higher probability than the current policy, suggesting that the decision is well supported by prior experience.

In contrast, when $\delta_t > 0$, the current policy assigns high confidence to a token that is strongly disfavored by the RL-optimized policy, exposing a clear mismatch with empirically validated decision patterns. Such overconfident yet unsupported decisions contradict RL experience and tend to arise at pivotal reasoning steps, where an incorrect choice can irreversibly divert the reasoning trajectory and propagate an incorrect final answer. We therefore regard these positions as critical decision points, where incorporating prior RL experience is most beneficial.

**Resampling.** Given the identified critical decision points, EAPO performs experience-guided action resampling during rollout generation. Concretely, we introduce a threshold $\tau$ to determine whether experience should be injected at the decoding step $t$:

$$g_t = \mathbb{I}(\delta_t > \tau), \tag{7}$$

where $g_t = 1$ indicates that step $t$ is identified as a critical decision point, and the candidate token $y_t$ is discarded and resampled from $\pi_{\mathrm{RL}}$. Based on this gating, we define the experience-guided sampling trajectory as:

$$\pi_{\mathrm{Exp}}(\cdot \mid q, y_{<t}) = \begin{cases} \pi_\theta(\cdot \mid q, y_{<t}), & \text{if } g_t = 0, \\ \pi_{\mathrm{RL}}(\cdot \mid q, y_{<t}), & \text{if } g_t = 1. \end{cases} \tag{8}$$

When $g_t = 0$, tokens are sampled directly from the current policy, preserving on-policy exploration. When $g_t = 1$, experience is injected by resampling the token from the prior RL-optimized policy. Importantly, EAPO does not replace entire trajectories; instead, it intervenes only at a small number of high-risk decision points, enabling fine-grained correction of the reasoning process while maintaining sufficient on-policy exploration.

**Acceleration via block-wise verification.** Applying experience verification at every decoding step can be computationally expensive. Inspired by speculative decoding (Chen et al., 2023; Leviathan et al., 2023), we amortize verification by processing $K$ decoding steps as a block.

At step $t$, the current policy $\pi_\theta$ generates a speculative block of $K$ tokens $\tilde{y}_{t:t+K-1}$. The RL policy $\pi_{\mathrm{RL}}$ is then applied in parallel to evaluate discrepancies $\{\delta_{t+i}\}_{i=0}^{K-1}$. If a critical point is detected, we identify the earliest index $i$ such that $g_{t+i} = 1$, truncate the sequence at $t + i$, and resample that token from $\pi_{\mathrm{RL}}$. All subsequent speculative tokens are discarded to preserve causal consistency. This block-wise verification substantially reduces the overhead of experience checking while preserving the original resampling behavior.

### 3.3. Experience-aware Policy Optimization

We next describe how experience-guided rollouts are incorporated into policy optimization in a stable manner.

**Positive Experience Filtering.** In EAPO, experience is introduced into only a small number of trajectories, typically a single experience-augmented response per rollout group. While experience-guided resampling often improves response quality, it does not guarantee correct outcomes for each rollout.

When an experience-augmented trajectory yields an incorrect prediction, the injected experience may be misaligned with the current policy, making gradient signals that deviate from the on-policy optimization objective. Such erroneous experience-augmented trajectories can therefore distort the policy update direction. In contrast, negative samples are more reliably obtained through on-policy exploration, which better reflects the model's current uncertainty and failure modes (Liu et al., 2025).

To mitigate this, we apply *positive experience filtering*, restricting policy optimization to experience-augmented trajectories that yield correct predictions. Formally, for the experience-augmented trajectory $o_{\mathrm{exp}}^b$, we define a binary mask $m_{\mathrm{exp}}^b$:

$$m_{\mathrm{exp}}^b = \begin{cases} 1, & \mathbf{R}(o_{\mathrm{exp}}^b, y^b) = 1, \\ 0, & \mathbf{R}(o_{\mathrm{exp}}^b, y^b) = 0, \end{cases} \tag{9}$$

ensuring that only trajectories with $m_{\mathrm{exp}}^b = 1$ are included in gradient updates.

**Resampling-based Importance Sampling.** Experience-guided resampling replaces a subset of tokens from $\pi_{\mathrm{old}}$ with a prior RL policy $\pi_{\mathrm{RL}}$. Consequently, experience-augmented responses are no longer sampled purely from $\pi_{\mathrm{old}}$, violating the standard importance sampling assumption (Shao et al., 2024; Schulman et al., 2017).

A naive approach would be to modify importance sampling at the token level. However, due to the auto-regressive nature of language models, token-wise correction breaks their causal dependency, since resampling a single token alters the effective distribution of all subsequent tokens.

Consequently, token-level importance sampling fails to characterize the behavior policy of the generated trajectory faithfully and often leads to unstable optimization.

Instead, we adopt a response-level, smoothed importance sampling scheme. For an experience-augmented response $o_i^b$, we compute the resampling ratio:

$$\rho_i^b = \frac{1}{|o_i^b|} \sum_{t=1}^{|o_i^b|} g_{i,t}^b, \tag{10}$$

which measures the proportion of tokens resampled from $\pi_{\mathrm{RL}}$. This statistic serves as a simple empirical proxy for the overall degree of experience injection in the trajectory. Based on this, we approximate the effective behavior policy $\pi_{\mathrm{Exp}}^b$ with a smoothed surrogate defined as:

$$\tilde{\pi}_{\mathrm{Exp}}^b = (1 - \rho_i^b)\,\pi_{\mathrm{old}} + \rho_i^b\,\pi_{\mathrm{RL}}. \tag{11}$$

And compute the final importance ratio as:

$$\tilde{r}_i^b = \frac{\pi_\theta(o_i^b \mid q^b)}{\tilde{\pi}_{\mathrm{Exp}}^b(o_i^b \mid q^b)}. \tag{12}$$

This formulation provides a smooth and stable correction for experience-guided rollouts. When no experience is injected ($\rho_i^b = 0$), it reduces to the standard importance ratio. As

the amount of experience increases, the correction smoothly interpolates toward the experience-guided behavior.

**Experience Annealing.** While experience-guided resampling accelerates early exploration, continued reliance on prior RL experience becomes unnecessary as the current policy improves. Once the policy has sufficiently benefited from experience guidance, further resampling may restrict exploration and hinder independent refinement.

EAPO therefore applies experience-guided resampling only for the first $T$ training steps and subsequently disables it. After annealing, the policy is trained purely with on-policy rollouts, enabling unrestricted exploration and continued improvement.

**Benefits.** EAPO offers three key advantages: **(1) Adaptability**: experience adapts naturally to prefixes generated by the current policy; **(2) Focus on critical decisions**: experience is injected only at high-impact decision points; and **(3) Balanced exploration**: sparse intervention preserves the on-policy nature of learning.

# 4. Experiment

In this section, we evaluate the effectiveness of our Experience-Augmented Policy Optimization over two base models: Qwen-2.5-Math-7B-Base (Yang et al., 2024b) and Qwen-3-8B-Base (Yang et al., 2025). Specifically, we aim to answer the following **R**esearch **Q**uestions (RQs):

❶: Is EAPO sensitive to hyperparameters?
❷: How do different components affect EAPO?
❸: How does EAPO compare to state-of-the-art methods?

## 4.1. Implementation Details

**Experimental Setup.** All experiments are conducted under the DAPO framework (Yu et al., 2025). Following the standard configuration, we set the clip-higher and clip-lower hyperparameters to $\epsilon_{\text{high}} = 0.28$ and $\epsilon_{\text{low}} = 0.2$, respectively. To support long-chain reasoning, the maximum response length is set to 20,480 tokens for Qwen3-8B-Base and 8,192 tokens for Qwen2.5-Math-7B.

In our setting, we adopt a two-stage training strategy. In the first stage, an RL-optimized reference policy $\pi_{\text{RL}}$ is obtained by optimizing the base model $\pi_{\text{Base}}$ with the DAPO objective. In the second stage, we initialize the target policy $\pi_\theta$ from the same base model $\pi_{\text{Base}}$ and further optimize it using DAPO, while selectively incorporating experience from $\pi_{\text{RL}}$. Importantly, $\pi_{\text{RL}}$ is only used to provide experience guidance during rollout and does not directly supervise or constrain the optimization objective.

**Training Dynamics.** All models are fine-tuned on the DAPO-Math-17K dataset (Yu et al., 2025). We use the

AdamW optimizer with a constant learning rate of $1 \times 10^{-6}$, a weight decay of $0.01$, and a global batch size of $512$. For each training step, we sample a rollout group of size $G = 16$. Within each group, $15$ responses are generated by the current policy $\pi_\theta$ to encourage on-policy exploration, while only $1$ response is experience-augmented trajectories guided by $\pi_{\text{RL}}$. For experience annealing, we set the block size to $K = 20$ and the annealing step to $T = 60$.

**Evaluation Protocols.** We evaluate our approach across two complementary dimensions. *Mathematical reasoning* benchmarks include AIME'24, AIME'25, and AMC. To assess general knowledge and reasoning ability beyond mathematics, we additionally report results on *general scientific knowledge* benchmarks, including MMLU-Pro (Wang et al., 2024) and GPQA (Rein et al., 2024).

## 4.2. Ablation Study

**Hyperparameter Sensitivity (RQ1).** We study the sensitivity of the resampling threshold $\tau$, which controls the trade-off between on-policy exploration and experience guidance. Figure 2 summarizes the effect of $\tau$ from three complementary perspectives: **(a)** AIME'24 performance, reflecting the model's capability in mathematical reasoning; **(b)** the resampled ratio, indicating the intensity of experience intervention during training; and **(c)** the resampled accuracy, which measures the quality of the injected experience.

• **Effects of Sparse Intervention.** A key observation from Figure 2 (b) and (c) is the high efficiency of experience injection. When $\tau = 0.5$, the resampled ratio remains extremely low (below 4%), indicating that EAPO intervenes at only a small subset of decision points (*e.g.,* approximately 40 tokens within a 1,000-token response). Despite this sparse intervention, the accuracy of experience-augmented responses consistently exceeds 70%. This *low-intervention, high-return* behavior substantiates our core hypothesis that effective RL guidance is inherently sparse, and that selectively correcting only a limited number of critical tokens is sufficient to reliably steer policy optimization.

• **Impact of the Threshold $\tau$.** The threshold $\tau$ determines the strictness of experience injection. For moderate values ($\tau \in [0.3, 0.5]$), although the resampled ratio gradually decreases as $\tau$ increases, both rollout accuracy and AIME'24 performance remain stable and high. This indicates that the most informative experience is preserved even under stricter filtering. In contrast, when $\tau$ is set too high ($\tau \geq 0.8$), the resampled ratio drops sharply, causing many critical decision points to be missed. As a result, both the resampled accuracy (Figure 2 (c)) and the task performance on AIME'24 (Figure 2 (a)) degrade noticeably.

• **Hyperparameter Setting.** To minimize reliance on $\pi_{\text{RL}}$ while maintaining strong performance, we adopt $\tau = 0.5$ as

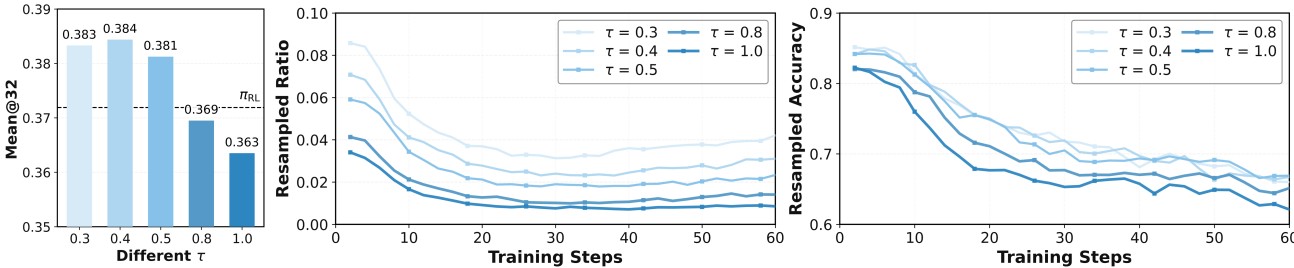

*Figure 2.* **Sensitivity analysis of the resampling threshold** $\tau$. (a) AIME'24 performance (Mean@32) under different values of $\tau$, reporting the best performance achieved during training. (b) Evolution of the resampled ratio across training steps, representing the proportion of resampled tokens within the experience-augmented responses. (c) Accuracy of experience-augmented responses across training steps, reflecting the quality of experience injection.

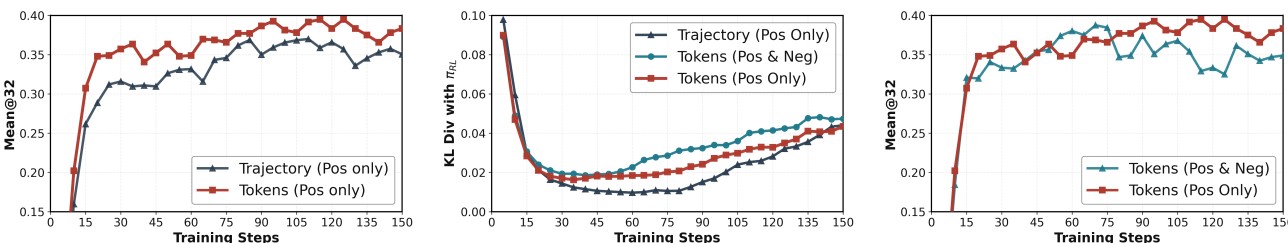

*Figure 3.* **Influence of experience granularity and filtering strategy.** (a) AIME'24 performance (Mean@32) comparison between token-level (Tokens, Pos-Only) and trajectory-level (Trajectory, Pos-Only) experience. Following (Zhan et al., 2025; Liu et al., 2025), we consider only positive samples in this comparison, where positive samples refer to trajectories whose final predictions are correct. (b) KL divergence to the RL-optimized experience model $\pi_{\text{RL}}$, measuring the distributional distance between the learned policy and the experience model under different experience designs. (c) AIME'24 performance (Mean@32) comparison between positive-only (Tokens, Pos-Only) and all (Tokens, Pos & Neg) token-level experience, illustrating the effect of different experience filtering strategies.

the default setting. This configuration achieves a favorable balance between sparse experience usage and effectiveness, reaching a Mean@32 score of 0.381 on AIME'24.

**Influences of Different Components (RQ2).** We analyze two key design choices in experience utilization: the granularity of experience (token-level versus trajectory-level) and the experience filtering strategy (positive-only versus using both positive and negative samples). Here, positive samples refer to trajectories whose final predictions are correct.

● **Effects of experience granularity.** Following (Liu et al., 2025; Zhan et al., 2025), we compare token-level and trajectory-level experience using only positive samples. As shown in Figure 3 (a), token-level experience consistently outperforms trajectory-level replay on AIME'24. Although trajectory-level experience yields the lowest KL divergence to the experience model $\pi_{\text{RL}}$ (Figure 3 (b)), it provides coarse-grained supervision that is insufficient for effective policy improvement. In contrast, token-level experience enables targeted correction at critical decision points, leading to better performance.

● **Effects of experience quality.** We further examine experience filtering by comparing positive-only token-level experience with the inclusion of both positive and negative samples. As shown in Figure 3 (c), incorporating negative

samples leads to a clear degradation in AIME'24 performance. This is accompanied by a larger KL divergence to $\pi_{\text{RL}}$ (Figure 3 (b)), suggesting that negative samples introduce conflicting optimization signals that interfere with effective experience utilization.

Overall, these results indicate that effective experience utilization requires both fine-grained intervention and careful filtering to maintain a moderate distance to the experience model, rather than indiscriminate replay.

### 4.3. Comparison with SOTA Methods (RQ3).

**Compared Methods and Categories.** To enable fair comparisons, we re-implement representative state-of-the-art RLVR methods and adapt them to the same training setting as EAPO. The compared methods can be broadly categorized into three groups.
**(1) Baseline RLVR methods**, include GRPO and DAPO, which optimize policies using outcome-level verifiable rewards without explicit experience reuse. Notably, DAPO also serves as the RL-optimized experience policy $\pi_{\text{RL}}$ in EAPO, highlighting that EAPO improves performance by reusing strong RLVR baselines.
**(2) Token-level supervision methods**, include on-policy distillation (OPD) and Multi-Teacher On-Policy Distillation

*Table 1.* **Performance on math reasoning benchmarks.** Comparison of RLVR strategies on in-domain math reasoning benchmarks, including AIME'24, AIME'25, and AMC. Due to time constraints and computational cost on the Qwen3-8B-Base model, Trajectory Replay and EAPO w/o sIS are only evaluated on the 7B models. Results are calculated over 32 runs, reported as proportions %.

| Model | Method | AIME'24 | | AIME'25 | | AMC | | Average | |
|---|---|---|---|---|---|---|---|---|---|
| | | Pass@1 | Pass@16 | Pass@1 | Pass@16 | Pass@1 | Pass@16 | Pass@1 | Pass@16 |
| Qwen-2.5 -Math-7B | GRPO | 32.08 | 50.24 | 11.77 | 24.56 | 67.39 | 81.24 | 37.08 | 52.01 |
| | DAPO | 37.19 | 58.00 | 15.62 | 30.60 | 68.86 | 89.59 | 40.56 | 59.40 |
| | MOPD | 38.85 | 60.28 | 15.93 | 33.08 | 68.56 | 87.67 | 41.11 | 60.34 |
| | OPD | 38.12 | 55.06 | 15.21 | 29.14 | 68.59 | 87.75 | 40.64 | 57.32 |
| | Trajectory Replay | 37.25 | 58.60 | 16.24 | 34.12 | 72.74 | **89.93** | 42.08 | 60.88 |
| | **EAPO w/o sIS** | 39.68 | **63.29** | 15.93 | 35.46 | 72.66 | 88.27 | 42.76 | 62.37 |
| | **EAPO** | **40.14** | 60.44 | **17.08** | **37.04** | **72.81** | 89.74 | **43.34** | **62.41** |
| Qwen-3 8B-Base | GRPO | 31.67 | 61.98 | 23.54 | 50.91 | 67.73 | 86.89 | 40.98 | 66.59 |
| | DAPO | 36.67 | 71.41 | 27.39 | 48.98 | 71.88 | 89.87 | 45.31 | 70.09 |
| | MOPD | 37.80 | **74.68** | 29.38 | 52.25 | 72.03 | 89.83 | 46.40 | 72.25 |
| | OPD | 38.23 | 67.48 | 27.81 | 50.90 | 71.16 | 88.17 | 45.73 | 68.85 |
| | **EAPO** | **41.66** | 74.37 | **30.92** | **52.54** | **76.54** | **91.51** | **49.71** | **72.81** |

*Table 2.* **Performance on general science benchmarks.** Comparison of different RLVR strategies on out-of-distribution reasoning benchmarks, including GPQA and MMLU-Pro. All results are averaged over 8 runs, reported as proportions (%).

| Model | Method | GPQA | MMLU-Pro | Avg |
|---|---|---|---|---|
| Qwen-2.5 -Math-7B | GRPO | 34.34 | 36.94 | 35.64 |
| | DAPO | 39.27 | 44.09 | 41.68 |
| | MOPD | 38.76 | 43.81 | 41.29 |
| | OPD | 39.39 | 43.34 | 41.37 |
| | Trajectory Replay | 39.67 | 46.50 | 43.09 |
| | **EAPO w/o sIS** | 41.98 | 47.35 | 44.67 |
| | **EAPO** | **43.18** | **47.57** | **45.38** |
| Qwen-3 8B-Base | GRPO | 50.44 | 64.71 | 57.58 |
| | DAPO | 52.27 | 65.29 | 58.78 |
| | MOPD | 50.51 | 65.99 | 58.25 |
| | OPD | 52.27 | 65.91 | 59.09 |
| | **EAPO** | **54.67** | **66.49** | **60.58** |

(MOPD), where an RL-optimized policy $\pi_{RL}$ is used to provide token-level guidance during optimization.

**(3) Trajectory-level experience replay methods**, include trajectory replay, where successful trajectories generated by $\pi_{RL}$ are reused as experience during training.

Moreover, we additionally report results for EAPO without smoothed importance sampling (denoted as EAPO w/o sIS), alongside the full EAPO model.

**Overall Performance.** Table 1 and Table 2 summarize the comparison between EAPO and representative RLVR baselines on in-domain math reasoning and out-of-distribution science benchmarks. Across all evaluated settings, EAPO consistently achieves the strongest or highly competitive performance, demonstrating its effectiveness as an experience-augmented policy optimization method.

● **In-domain Math Reasoning.** On math reasoning bench-marks (Table 1), EAPO consistently outperforms baseline RLVR methods (GRPO, DAPO) and token-level supervision approaches (OPD, MOPD) across model scales. Compared with trajectory-level experience replay, EAPO achieves higher average performance, indicating that selectively injecting experience at critical decision points is more effective than replaying entire trajectories.

● **Out-of-Domain Science Reasoning.** EAPO also demonstrates strong generalization on out-of-distribution science benchmarks (Table 2). On both GPQA and MMLU-Pro, EAPO consistently surpasses all compared baselines under the same training setting, suggesting that it learns transferable reasoning behaviors rather than overfitting to domain-specific patterns.

● **Effect of Importance Sampling.** The consistent performance gap between EAPO and its variant without smoothed importance sampling (EAPO w/o sIS) highlights the importance of stable credit assignment when incorporating experience. Smoothed importance sampling regulates gradient contributions from experience-augmented decisions, leading to more stable optimization even when only a small fraction of tokens are resampled by $\pi_{RL}$.

Overall, EAPO achieves superior performance by combining fine-grained experience injection with careful optimization control, providing a more effective alternative over existing RLVR paradigms.

## 5. Related Works

In this section, we first briefly summarize current RLVR strategies, and then we illustrate related methods for enhancing large language models via experience. Finally, we enumerate the differences.

## 5.1. Reinforcement Learning with Verifiable Rewards.

Large language models and its variances have developed rapidly in recent years (Lu et al., 2023; 2024; 2025; 2026a; Li et al., 2025a; Yang et al., 2026), and more recently, Reinforcement Learning with Verifiable Rewards (RLVR) has further advanced their reasoning capabilities (Guo et al., 2025; Lu et al., 2026c; Li et al., 2026; Lu et al., 2026b). Specifically, pioneering works, such as OpenAI's o1 (Jaech et al., 2024) and DeepSeek-R1 (Guo et al., 2025), have established the paradigm that RLVR can significantly enhance the reasoning capabilities of LLMs. Building on this, recent state-of-the-art models (Team et al., 2025; Yang et al., 2025; Liu et al., 2025) further push the boundaries across diverse and complex scenarios. Technically, RLVR leverages rule-based verifiers to provide precise, outcome-based rewards for tasks with concise ground truths, such as mathematics and programming. Representative algorithms, such as Group Relative Policy Optimization (GRPO) (Shao et al., 2024) and its successor, Dynamic Sampling Policy Optimization (DAPO) (Yu et al., 2025), have established themselves as foundational baselines, inspiring a wide range of follow-up studies (Wu et al., 2025; Cheng et al., 2025; Wang et al., 2025). However, we observe that most existing RLVR frameworks rely heavily on pure on-policy sampling, often neglecting the potential of expert experience to guide the optimization process. This motivates our investigation into a more robust experience injection strategy.

## 5.2. Off-policy Guidance for RLVR.

To mitigate the high sampling inefficiency and limited exploration of pure on-policy RLVR, recent work has explored incorporating off-policy guidance during RL rollout. Existing approaches can be broadly categorized into two categories.

**Distillation from stronger models.** One line of work attributes the exploration bottleneck to insufficient model capacity and addresses it by distilling stronger models into the current model. Representative approaches, such as (Yan et al., 2025; Lu & Lab, 2025; Jiang et al., 2025; Zhang et al., 2025c), employ more capable models (*e.g.,* Qwen-2.5 32B(Yang et al., 2024a), DeepSeek-R1 (Guo et al., 2025)) to provide high-quality trajectories or token-level guidance, thereby improving the model's reasoning performance.

**Experience reuse from previous success.** Another line of work focuses on reusing experience accumulated from the model's own reinforcement learning process. Representative methods incorporate successful experiences from earlier RL iterations or optimized policies into current rollouts (Zhan et al., 2025; Li et al., 2025b). Our work belongs to the experience-based category. Different from distillation-based approaches, we believe that the base model already possesses sufficient reasoning capacity, and instead aim to unlock its potential by more effectively exploiting experi-

ence accumulated through prior RL optimization.

## 5.3. Experience in Large Language Models.

As we enter the *era of experience* (Silver & Sutton, 2025), the focus of reinforcement learning for LLMs is shifting from pure online exploration toward the strategic utilization of experiential knowledge. In agent learning, leveraging early or synthesized experience has been shown to significantly bolster generalization capability across novel domains (Zhao et al., 2024; Zhang et al., 2025b). Similarly, in large reasoning models, recent approaches incorporate experience primarily at the trajectory level. Representative strategies either reuse trajectories from optimized policies (Zhang et al., 2025a) or revisit rollouts from previous on-policy iterations (Zhan et al., 2025; Li et al., 2025b; Liang et al., 2025). While these methods prove that replaying successful trajectories accelerates convergence, they typically treat experience as holistic, trajectory-level demonstrations, overlooking the sparsity of critical decisions within reasoning processes, and may suffer from trajectory mismatch as the policy evolves. In contrast, EAPO models experience at the action level and injects it at critical decision points, enabling policy-adaptive experience reuse while avoiding trajectory-level mismatch.

## 6. Conclusion

In this work, we identify a fundamental limitation of existing RLVR methods: they primarily rely on on-policy optimization from scratch and fail to effectively reuse the rich experience accumulated in prior RL-optimized models. To address this limitation, we propose **E**xperience-**A**ugmented **P**olicy **O**ptimization (EAPO), which represents RL experience as an action-level, policy-adaptive prior rather than static trajectories. EAPO selectively injects experience at critical decision points during rollout and integrates experience-augmented trajectories into on-policy optimization through an experience-aware optimization strategy.

Extensive experiments on both mathematical and general reasoning benchmarks demonstrate that EAPO consistently improves reasoning performance and generalization while requiring only sparse experience intervention. Overall, these results suggest that policy-adaptive experience reuse offers an effective and scalable direction for enhancing reinforcement learning with verifiable rewards.

**Limitations and Future Work** Despite its effectiveness, EAPO has several limitations that leave room for future research: **(1)** EAPO relies on a prior RL-optimized policy $\pi_{RL}$, and its effectiveness is bounded by the quality and diversity of this policy. Extending EAPO to multiple or dynamically evolving experience policies is a natural next step. **(2)** EAPO employs predefined mechanisms, such as a

fixed resampling threshold and annealing schedule. More adaptive strategies that adjust experience usage based on training dynamics may further improve robustness. **(3)** Our study focuses on reasoning tasks with verifiable rewards; extending experience-guided resampling to weaker or noisier supervision remains an open question.

## Impact Statement

This paper presents work whose goal is to advance the field of machine learning. There are many potential societal consequences of our work, none of which we feel must be specifically highlighted here.

## Acknowledgement

This research is supported by the National Natural Science Foundation of China (62572449).

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

## A. More Ablation Results.

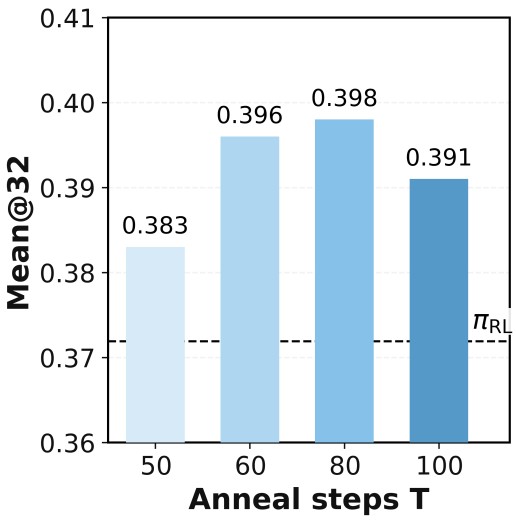
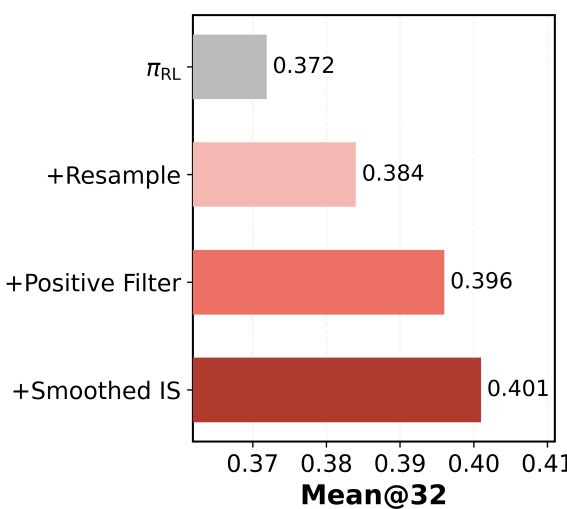

*(a)* **Effect of experience annealing steps.** AIME'24 performance under different annealing steps $T$. Insufficient annealing leads to slightly lower gains.

*(b)* **Component-wise ablation of EAPO.** Adding resampling, positive filtering, and smoothed importance sampling yields monotonic AIME'24 performance gains.

*Figure 4.* **Additional ablation results.** Left: impact of experience annealing steps. Right: complementary effects of EAPO components.

**Effect of experience annealing.** Figure 4a studies the impact of the experience annealing steps $T$. Across all settings, incorporating experience consistently improves performance over the baseline, confirming the general benefit of experience-guided resampling. When the annealing period is too short, insufficient exposure to experience limits its effectiveness, whereas extending experience usage excessively yields diminishing returns as the policy becomes sufficiently optimized. We therefore set $T = 60$ to leverage experience in the early training stage while avoiding unnecessary intervention for later on-policy refinement.

**Component-wise analysis.** Figure 4b presents a step-by-step ablation starting from the RL-optimized policy $\pi_{\mathrm{RL}}$. Introducing experience-guided resampling to $\pi_\theta$ already yields a clear performance gain over $\pi_{\mathrm{RL}}$, indicating that injecting action-level experience during rollout is effective and can further improve upon the experience policy itself. Adding positive experience filtering leads to additional gains by stabilizing optimization and mitigating misleading gradients from incorrect experience. Finally, smoothed importance sampling provides a further boost, highlighting the importance of proper credit assignment when integrating experience-augmented trajectories. Together, these results demonstrate that the components of EAPO are complementary and jointly contribute to its overall performance improvements.

# B. Algorithm of EAPO.

---

**Algorithm 1** Experience-Augmented Policy Optimization (EAPO)

---

**Require:** Current policy $\pi_\theta$, Prior RL policy $\pi_{\text{RL}}$, Group size $G$, Threshold $\tau$, Block size $K$, Annealing step $T$.

1: Initialize $\pi_{\text{old}} \leftarrow \pi_\theta$.
2: **for** each training step $s = 1, 2, \ldots$ **do**
3:     // Phase 1: Rollout (On-policy Rollout + Experience-augmented Rollout)
4:     Sample a batch of questions $\{q^b\}_{b=1}^B \sim \mathcal{D}$.
5:     **for** each question $q^b$ in parallel **do**
6:         Sample $G - 1$ on-policy trajectories $\mathcal{O}_{\text{on}}^b = \{\boldsymbol{o}_1^b, \ldots, \boldsymbol{o}_{G-1}^b\}$ from $\pi_{\text{old}}$.
7:         **if** $s \leq T$ **then**
8:             Generate experience-augmented trajectory $\boldsymbol{o}_{\text{exp}}^b$ via **Accelerated Block-wise Resampling**:
9:             **while** not EOS **do**
10:                 Generate speculative block $\tilde{y}_{t:t+K-1}^b \sim \pi_{\text{old}}$.
11:                 Compute discrepancies $\{\delta_{t+i}^b\}_{i=0}^{K-1}$ by comparing $\pi_{\text{old}}$ and $\pi_{\text{RL}}$.
12:                 **if** $\exists i$ s.t. $\delta_{t+i}^b > \tau$ **then**
13:                     Find $i^* = \min\{i \mid \delta_{t+i}^b > \tau\}$, resample $y_{t+i^*}^b \sim \pi_{\text{RL}}$, and set $g_{t+i^*}^b = 1$.
14:                     Truncate and update prefix $y_{<t+i^*+1}^b$.
15:                 **else**
16:                     Accept block; set $g_{t:t+K-1}^b = 0$.
17:                 **end if**
18:             **end while**
19:             Collect group $\mathcal{G}^b = \mathcal{O}_{\text{on}}^b \cup \{\boldsymbol{o}_{\text{exp}}^b\}$.
20:         **else**
21:             Sample $G$ trajectories from $\pi_{\text{old}}$ to form $\mathcal{G}^b$ (Experience Annealing).
22:         **end if**
23:     **end for**
24:     // Phase 2: Experience-aware Batched Optimization
25:     **for** each group $\mathcal{G}^b$ **do**
26:         Compute rewards $\{R_i^b\}_{i=1}^G$ and group-relative advantages $A_i^b = \frac{R_i^b - \text{mean}(R^b)}{\text{std}(R^b)}$.
27:         **for** each trajectory $\boldsymbol{o}_i^b \in \mathcal{G}^b$ **do**
28:             **if** $\boldsymbol{o}_i^b$ is the experience-augmented response $\boldsymbol{o}_{\text{exp}}^b$ **then**
29:                 **if** $\text{R}(\boldsymbol{o}_i^b, y^b) = 0$ **then**
30:                     **continue** {// Positive Filtering: skip failed experience}
31:                 **end if**
32:                 Compute $\rho_i^b = \frac{1}{|\boldsymbol{o}_i^b|} \sum_t g_{i,t}^b$ and $\tilde{\pi}_{\text{Exp}}^b = (1 - \rho_i^b)\pi_{\text{old}} + \rho_i^b \pi_{\text{RL}}$.
33:                 $r_i^b(\theta) \leftarrow \pi_\theta(\boldsymbol{o}_i^b \mid q^b)/\tilde{\pi}_{\text{Exp}}^b(\boldsymbol{o}_i^b \mid q^b)$.
34:             **else**
35:                 $r_i^b(\theta) \leftarrow \pi_\theta(\boldsymbol{o}_i^b \mid q^b)/\pi_{\text{old}}(\boldsymbol{o}_i^b \mid q^b)$ {// Standard on-policy ratio}
36:             **end if**
37:         **end for**
38:     **end for**
39:     Update $\theta$ by maximizing $\mathcal{J}_{\text{DAPO}}(\theta) = \mathbb{E}_{b,i}\left[\min(r_i^b \cdot A_i^b, \text{clip}(r_i^b) \cdot A_i^b)\right]$.
40:     $\pi_{\text{old}} \leftarrow \pi_\theta$.
41: **end for**

---

