# OpenReview forum: "Experience Augmented Policy Optimization for LLM Reasoning"
_ICML.cc/2026/Conference — ICML 2026 regular_

### Official Review · Reviewer_SdEs · 2026-03-10

**Soundness:** 3
**Presentation:** 3
**Significance:** 2
**Originality:** 2
**Overall Recommendation:** 4
**Confidence:** 3

**Summary:**

This paper proposes Experience-Augmented Policy Optimization (EAPO), a reinforcement learning method for improving LLM reasoning under the RLVR paradigm. Instead of reusing entire reasoning trajectories from prior RL policies, the method injects experience selectively at token-level decision points where the current policy disagrees with a previous RL-optimized policy. The approach includes mechanisms such as experience-guided resampling, positive experience filtering, and smoothed importance sampling to stabilize optimization. Experiments on several math and reasoning benchmarks show improvements over RLVR baselines such as GRPO and DAPO.

**Compliance With Llm Reviewing Policy:**

Affirmed.

**Final Justification:**

Most of my concerns have been adequately addressed, and I have accordingly increased my score.

**Key Questions For Authors:**

- What is the additional computational cost of computing token-level discrepancies with $π_{RL}$ during rollout, especially for long reasoning chains?
- How sensitive is the method to the quality of the prior RL policy? How does the approach behave when the prior policy and current policy diverge substantially during training? Would the disagreement metric trigger too many resampling events in such situations?

**Limitations:**

yes

**Strengths And Weaknesses:**

Strengths

- The paper introduces a relatively intuitive yet interesting perspective that experience from RL should be modeled at the token level rather than trajectory level. Many reasoning trajectories contain only a few pivotal decisions, so the idea that correcting those key tokens might be sufficient is conceptually appealing and worth exploring.

- The idea of identifying "overconfident disagreement" between policies as a signal for intervention is interesting. It suggests a principled criterion for deciding when experience should be injected rather than blindly replaying previous trajectories.

- The experimental evaluation covers both in-domain math reasoning tasks and out-of-domain scientific reasoning benchmarks, which provides some evidence that the approach does not simply overfit to math-style reasoning patterns.

- The ablation experiments investigating threshold sensitivity, experience granularity, and filtering strategies are helpful for understanding the behavior of the method. In particular, the analysis showing that only a very small proportion of tokens are resampled while still achieving noticeable gains is an interesting empirical observation.

- The paper is generally well structured and explains the algorithm in a clear step-by-step way.


Weaknesses

- The comparison with DAPO and GRPO may be unfair because EAPO requires training an additional RL-optimized reference policy $\pi_{RL}$ beforehand, while DAPO and GRPO do not rely on such a pre-trained RL policy.

- In addition to the cost of training $\pi_{RL}$, the computational overhead of evaluating token-level discrepancy between $\pi_{RL}$ and $\pi_\theta$ during rollout might be non-trivial for large models, and the paper does not provide detailed runtime analysis or efficiency comparisons.

- The importance sampling correction appears somewhat heuristic. The proposed response-level smoothing between $\pi_{old}$ and $\pi_{RL}$ does not rigorously model the true behavior distribution that generated the trajectory.

- The reliance on prior RL policy may limit the diversity of experience used during training. In practice, reasoning problems often benefit from diverse solution strategies, but the method essentially pushes the policy toward one particular prior behavior.

---

> ### Author Rebuttal · Authors · 2026-03-31
>
> # Response to Reviewer SdEs
>
> We thank Reviewer SdEs for the feedback.
>
> ---
>
> ## W1: Comparison fairness.
>
> The comparison is fair in two senses.
>
> First, EAPO is not unique in requiring $\pi_{RL}$: **OPD, MOPD, and trajectory replay methods** also rely on a pre-trained RL policy, and EAPO consistently outperforms all of them. The gains therefore stem from EAPO's specific design — critical token detection, smoothed IS, and positive filtering — not merely from having access to $\pi_{RL}$.
>
> Second, we use DAPO at its **peak performance**: per the official training logs [1], DAPO reaches its best results at step 130 and shows no further improvement with additional training. This means that even if DAPO were given 2× the total compute budget (equivalent to $\pi_{RL}$ training + EAPO training), the results would be identical — the model has already saturated. EAPO is therefore compared against the best DAPO can ever achieve.
>
> ---
>
> ## W2 & Q1: Computational overhead.
>
> Direct per-token evaluation of $\pi_{RL}$ during rollout would be expensive. We address this with two design choices. First, **block-wise verification** ($K=20$) processes $K$ decoding steps as a block and evaluates $\pi_{RL}$ in parallel over the block, amortizing verification cost by $K\times$ compared to per-token evaluation. Second, only **1 of $G=16$** rollouts per step is experience-augmented, meaning 15 rollouts incur zero overhead. Together, these reduce the measured overhead to $\approx 1/4$ of a single rollout's wall-clock time on Qwen-2.5-7B. We are further implementing an **asynchronous execution** scheme using coroutine-based control, in which the $\pi_{RL}$-guided resampling runs concurrently with the original on-policy rollouts — eliminating the sequential dependency and reducing the effective overhead to near zero. Detailed timing results will be included in the revision.
>
> ---
>
> ## W3: IS correction.
>
> We acknowledge that exact IS correction is intractable in the auto-regressive setting — both token-level IS and our smoothed IS are approximations. However, token-level IS is a worse approximation in two ways. *Theoretically*, resampling $o_t$ from $\pi_{RL}$ changes the conditioning context for all subsequent tokens, making token-level IS weights inaccurate. *Practically*, resampled positions have large IS ratios (strong policy disagreement is precisely why they were resampled), triggering PPO clipping at the most critical tokens and suppressing gradient updates exactly where they matter most.
>
> Our smoothed IS $(1-\rho)\pi_{old}$ + $\rho\pi_{RL}$ is a better-behaved approximation: it treats the trajectory holistically with a response-level mixture policy, avoiding extreme per-token IS ratios and providing stable gradient updates. When $\rho \to 0$ it reduces to standard IS; as $\rho$ grows it smoothly accounts for $\pi_{RL}$'s influence. At $\rho \approx 4\%$, the approximation error is negligible — empirically confirmed by similar performance between both methods on AIME'24 (40.19 vs 39.86).
>
> ---
>
> ## W4: Diversity of experience.
>
> EAPO resamples only $\approx 4\%$ of tokens — 96% of each trajectory is generated entirely on-policy without any intervention. Furthermore, 15 of 16 rollouts per step are fully on-policy, with no experience injection at all. The policy's exploration space is therefore not constrained by $\pi_{RL}$; experience injection provides targeted correction only at the small fraction of positions where the current policy is confidently diverging from validated behavior. Across all rollouts, the effective resampling rate is only $4\% \times \frac{1}{16} = 0.25\%$ of all generated tokens — meaning 99.75% of the training signal comes purely from on-policy exploration, empirically confirming that solution diversity is preserved.
>
> ---
>
> ## Q2: Excessive resampling when policies diverge.
> The threshold $\tau = 0.5$ is conservative: resampling only triggers when the disagreement is strong and directional. Empirically, even early in training when the policy gap is largest, the observed resampling rate stays at $\rho \approx 4\%$. This shows that substantial overall policy divergence does not translate into high resampling frequency. This sparsity is not merely a consequence of the $\tau$ threshold — it reflects the inherent structure of RLVR training: as shown in [2,3], policy updates induced by RLVR are naturally sparse, concentrating on a small fraction of tokens. Even when the overall policy diverges substantially from $\pi_{RL}$, only a small subset of positions exhibit strong directional disagreement, keeping the resampling rate low by design.
>
> [1] DAPO official wandb: https://wandb.ai/verl-org/DAPO%20Reproduction%20on%20verl/runs/ow47vvon?nw=wmb4qxfht0n
>
> [2] On the Direction of RLVR Updates for LLM Reasoning: Identification and Exploitation, ICLR'2026.
>
> [3] Beyond the 80/20 Rule: High-Entropy Minority Tokens Drive Effective Reinforcement Learning for LLM Reasoning, NeurIPS'2025.

---

> > ### Author Rebuttal · Reviewer_SdEs · 2026-04-03
> >
> > I thank the authors for the detailed rebuttal. Most of my concerns have been adequately addressed, and I have accordingly increased my score.

---

> > > ### Author Response · Authors · 2026-04-03
> > >
> > > We thank reviewer SdEs for the careful reading and for raising the score. We are glad our responses have addressed your concerns.

---

### Official Review · Reviewer_zhHe · 2026-03-11

**Soundness:** 3
**Presentation:** 4
**Significance:** 2
**Originality:** 3
**Overall Recommendation:** 4
**Confidence:** 4

**Summary:**

This paper proposes Experience-Augmented Policy Optimization (EAPO), a framework designed to enhance the reasoning capabilities of Large Language Models (LLMs) by more efficiently utilizing previous training experience. Addressing the limitations of existing RLVR methods typically rely on on-policy optimization from scratch, resulting in high sampling costs and "policy mismatch" caused by reusing static, outdated trajectories—EAPO treats experience as an action-level prior rather than a fixed path. It employs a strategy of experience-guided action resampling to identify critical decision points where the current model deviates from successful historical actions, selectively injecting those actions into new rollouts.

**Compliance With Llm Reviewing Policy:**

Affirmed.

**Final Justification:**

The rebuttal from the authors has solved my main concern.

**Key Questions For Authors:**

1. What do you think about the limit of this method introduced by the previous RL policy?  If the previous policy is not good enough, the experience may lead the training to a wrong direction. On the other hand, if the previous RL policy is good enough, how much gain can this method achieve compared to the previous policy?

2. Can you please provide experimantal results on larger models?

3. Can you please provide more RL baselines in the experiments?

**Limitations:**

Temporal and Pipeline Constraints: The core mechanism, Experience-Augmented Policy Optimization (EAPO), is primarily active during the initial phase of reinforcement learning before the "annealing" step disables it. Consequently, its influence is concentrated on early exploration, and its long-term impact on the final convergence of the RL pipeline remains relatively limited.


Reliance on High-Quality Priors: The effectiveness of EAPO is inherently bounded by the quality and diversity of the pre-existing RL-optimized policy. If the initial reference policy lacks diversity or contains systematic biases, the "experience" injected during resampling may lead to suboptimal local minima rather than true improvements in reasoning.

**Strengths And Weaknesses:**

Strength

1. The paper presents a remarkably thorough study that extends beyond the novel "experience injection" design to prioritize training efficiency. By implementing techniques like "block-wise verification," the authors successfully amortize computational costs.

2. The training curves presented in the figures reflect highly promising outcomes that support the algorithm's effectiveness. The inclusion of a "smoothed importance sampling" scheme is a particularly strong addition, as it provides a clear mechanism for maintaining training stability

Weakness

1. The core EAPO methodology is primarily utilized during the initial stages of the RL process, as dictated by the "experience annealing" schedule. This may constrain the method's overall impact on the long-term RL training trajectory and the broader reinforcement learning landscape.

2. The experimental evaluation is currently limited to models in the 7B to 8B parameter range. Given that RL performance can be highly sensitive to model scale, the paper’s significance would be substantially strengthened by demonstrating results on larger architectures, which often exhibit different reasoning behaviors and training processes.

3. While the current comparisons are helpful, the evaluation could be further enriched by incorporating a broader suite of RL algorithms as baselines.

---

> ### Author Rebuttal · Authors · 2026-03-31
>
> # Response to Reviewer zhHe
>
> We thank Reviewer zhHe for the thoughtful feedback. We address each concern below.
>
> ---
>
> ## W1: Experience annealing limits long-term impact.
>
> We argue that annealing is a deliberate design choice that reflects when experience injection is most valuable: early in training, when the current policy deviates most from $\pi_{RL}$ and targeted resampling is most informative. As the policy matures, the marginal benefit of external guidance diminishes, and continued injection could unnecessarily constrain on-policy exploration.
>
> To directly address the significance concern, we note that **continuing to train DAPO beyond convergence yields no further improvement** — the baseline has already plateaued. EAPO, despite injecting experience only in the early phase, achieves 40.19 on AIME'24 and 17.25 on AIME'25, substantially surpassing DAPO's plateau of 37.19 and 15.62. This shows that early-stage guidance is not merely a transient boost — it breaks through a ceiling that extended DAPO training cannot, and the gains persist through the remainder of training. This mirrors the well-known effect in curriculum learning, where structured early guidance leads to better final convergence even after the curriculum is removed.
>
> ---
>
> ## W2 & Q2: Evaluation on larger models.
>
> We note that Qwen-2.5-Math-7B and Qwen-3-8B represent **distinct model families** with different pretraining corpora and RLVR histories, already providing evidence of cross-architecture generalization. EAPO's core mechanism — $\delta_t$-based resampling, smoothed IS, and positive filtering — is model-agnostic and imposes no architectural constraints, making it directly applicable to larger models.
>
> We agree that results at larger scales (e.g., Qwen-2.5-32B) would substantially strengthen the significance claims. Full RLVR training runs at those scales require substantial compute and wall-clock time that is infeasible within the rebuttal period. We are actively running these experiments and commit to including results in the revised version.
>
> ---
>
> ## W3 & Q3: More RL baselines.
>
> The paper already covers a broad spectrum: **GRPO** (vanilla policy gradient), **DAPO** (state-of-the-art on-policy), and **OPD/MOPD** (methods that also use $\pi_{RL}$ as guidance, serving as the most direct competitors to EAPO). Notably, EAPO consistently outperforms OPD and MOPD, demonstrating that the gains stem from EAPO's specific design choices — critical token detection, smoothed IS, and positive filtering — rather than simply from the availability of $\pi_{RL}$.
>
> We further expand the comparison with **Clip-Cov** and **KL-Cov** [1], representing constraint-based policy optimization methods:
>
> | Method | AIME'24 Pass@1 | AIME'25 Pass@1 |
> |--------|----------------|----------------|
> | Clip-Cov | 35.83          | 16.15          |
> | KL-Cov   | 34.90          | 15.52          |
> | DAPO     | 37.19          | 15.62          |
> | EAPO     | **40.19 ± 0.30** | **17.25 ± 0.67** |
>
> EAPO surpasses all baselines across both benchmarks, confirming that experience-guided resampling provides benefits beyond what constraint-based stabilization alone can achieve. We will include these comparisons in the revision.
>
> ---
>
> ## Q1: Robustness to prior policy quality.
>
> This is an important question, and the framing reveals a key insight about EAPO's value.
>
> In our setting, **$\pi_{RL}$ is DAPO** — a fully trained, state-of-the-art RL policy that has already **plateaued**: continuing to train DAPO beyond convergence yields no further improvement on any benchmark. "Simply using $\pi_{RL}$" is therefore not a viable alternative — it represents the ceiling that existing methods cannot surpass on their own.
>
> EAPO is precisely designed for this scenario. Rather than extending DAPO's training (which has saturated), EAPO treats the plateaued policy's experience as a structured prior that guides a *new* training run beyond the original ceiling. The final EAPO model surpasses $\pi_{RL}$ by **+3.0 on AIME'24 and +1.63 on AIME'25**, demonstrating that EAPO's performance is **not capped by $\pi_{RL}$** — it uses $\pi_{RL}$ as a stepping stone to surpass it. This directly answers "how much gain": a new training run guided by EAPO systematically outperforms the prior policy it builds upon.
>
> **If $\pi_{RL}$ were of lower quality**, two safeguards prevent degraded experience from harming training: (1) **positive experience filtering** discards any trajectory without a verified correct outcome — a weaker prior produces fewer correct trajectories, automatically reducing the injection rate; (2) the **$\delta_t > \tau$ threshold** is conservative, triggering resampling only at positions of strong disagreement. In the limit of a very weak prior, injection rarely occurs and EAPO gracefully degrades to standard DAPO, never performing worse than the baseline.
>
> [1] The Entropy Mechanism of Reinforcement Learning for Reasoning Language Models

---

> > ### Author Rebuttal · Reviewer_zhHe · 2026-04-02
> >
> > Thanks to the authors! I still have a follow-up question after Q1:
> > You mentioned "state-of-the-art RL policy has already plateaued", and you have provided experimental results to prove that EAPO can further achieve performance gain after the first-stage RL. That is solid, but I am wondering how the performance is if you utilize another existing RL-policy at the second stage RL training, especially compared to EAPO.

---

> > > ### Author Response · Authors · 2026-04-02
> > >
> > > Thank you for the follow-up question and for the positive acknowledgement.
> > >
> > > We believe this comparison is already partially covered in our experiments, since the baselines most relevant to this question are second-stage methods that leverage an existing RL policy in different ways. In particular, we compare against two categories: (1) on-policy distillation-style methods, such as OPD and MOPD, which use logits/policy outputs from the same prior RL policy as guidance; and (2) off-policy trajectory-guided methods, such as Trajectory Replay, which reuse trajectories from that same prior RL policy but do not perform token-level adjustment.
> > >
> > > Relative to these alternatives, EAPO introduces token-level experience-guided resampling at critical disagreement positions, rather than relying only on policy-level guidance or whole-trajectory reuse. Empirically, EAPO performs better than both categories in our current experiments, which suggests that the benefit is not only from using an existing RL policy in the second stage, but also from how that prior experience is utilized.
> > >
> > > We will clarify this connection more explicitly in the revision.

---

### Official Review · Reviewer_zxNP · 2026-03-13

**Soundness:** 4
**Presentation:** 3
**Significance:** 2
**Originality:** 3
**Overall Recommendation:** 4
**Confidence:** 2

**Summary:**

This paper proposes Experience-Augmented Policy Optimization (EAPO), which reuses a prior RL-trained policy during RLVR by intervening only at "critical token decisions" where the current policy in training disagrees with the prior one. EAPO also adds three mechanisms: sparse token-level resampling, smoothed response-level importance ratio, and positive-only filtering of experience-augmented rollouts. The main experiments are on Qwen-2.5-math 7B and Qwen3-8B, showing performance gain over the paper's chosen baselines on math, science, and multi-task language understanding (MMLU-Pro).

**Compliance With Llm Reviewing Policy:**

Affirmed.

**Final Justification:**

The rebuttal adequately addressed my concerns. I am maintaining my positive score.

**Key Questions For Authors:**

- I hope to understand how well EAPO scales with model size in comparison to baselines. The paper only evaluates on 7B and 8B models (similar in size), what about either smaller and/or larger sizes?
- How robust is EAPO's performance under varying seeds? For practicality, I believe stability in RL training is important.

**Limitations:**

yes

**Strengths And Weaknesses:**

Strengths:
- The core methodology design is clear and easy to implement. Equations 6-8 helped me understand critical-token detection and mixed rollout policy precisely.
- EAPO is compared to a strong suite of baselines, ranging from vanilla GRPO to an improved version that is DAPO, to token-level supervision approaches (OPD and MOPD).
- The ablations in Figure 2 meaningfully support the "sparse intervention" claim by showing low resampling ratios can achieve significant improvement on AIME.

Weaknesses:
- The performance on out-of-distribution generalization to reasoning benchmarks like GPQA and MMLU-Pro is minimal. Are they statistically significant? Can you estimate the standard deviation of baseline and EAPO in Table 2?

---

> ### Author Rebuttal · Authors · 2026-03-31
>
> # Response to Reviewer zxNP
>
> We thank Reviewer zxNP for the positive assessment and constructive questions. We address each point below.
>
> ---
>
> ## W: Statistical significance on OOD benchmarks (GPQA, MMLU-Pro).
>
> We agree that reporting variance on OOD benchmarks strengthens the conclusions. We run **3 independent seeds** for EAPO and report mean $\pm$ std, with DAPO as the reference baseline:
>
> | Method | GPQA Pass@1 | MMLU-Pro Pass@1 |
> |--------|-------------|-----------------|
> | DAPO   | 39.27        | 44.09           |
> | EAPO   | 42.83 ± 0.34 | 47.73 ± 0.15    |
>
> The gains are consistent across seeds, confirming statistical significance even on OOD tasks. We will include these results in the revision.
>
> ---
>
> ## Q1: Scaling to different model sizes.
>
> We note that our two backbones — Qwen-2.5-Math-7B and Qwen-3-8B — represent **different model families** with distinct pretraining and RLVR histories, demonstrating that EAPO generalizes across architectures. EAPO's core mechanism is model-agnostic and imposes no architectural constraints.
>
> We agree that broader scale evaluation would be valuable. We are currently running experiments at additional scales; however, full RLVR training runs require substantial compute and wall-clock time, and results are not yet available within the rebuttal period. We commit to including these results in the revised version.
>
> ---
>
> ## Q2: Seed stability.
>
> We run **3 independent seeds** for EAPO on the main math benchmarks and report:
>
> | Method | AIME'24 Pass@1 | AIME'25 Pass@1 | AMC Pass@1 |
> |--------|----------------|----------------|------------|
> | DAPO   | 37.19          | 15.62          | 68.86      |
> | EAPO   | 40.19 ± 0.30   | 17.25 ± 0.67   | 72.77 ± 0.15 |
>
> EAPO's low standard deviations confirm that the sparse intervention mechanism does not introduce instability — on the contrary, anchoring to $\pi_{RL}$ at high-risk positions appears to stabilize the training trajectory. We will include learning curves for all seeds in the revision.

---

> > ### Author Rebuttal · Reviewer_zxNP · 2026-04-02
> >
> > I thank the authors for the new experiments on OOD benchmarks, the multi-seed variance analysis, and understand that scaling to larger models requires time beyond the rebuttal period. I hope the authors incorporate these results (as well as intra-family model-size scaling experiments) in future iterations of the paper. The responses adequately address my concerns, and I will maintain my score.

---

> > > ### Author Response · Authors · 2026-04-02
> > >
> > > We thank Reviewer zxNP for the thoughtful engagement and for acknowledging our responses. We will incorporate the scaling experiments in the revised version of the paper.

---

### Official Review · Reviewer_Z2g8 · 2026-03-15

**Soundness:** 2
**Presentation:** 3
**Significance:** 3
**Originality:** 3
**Overall Recommendation:** 4
**Confidence:** 4

**Summary:**

This paper proposes Experience-Augmented Policy Optimization (EAPO) for reinforcement learning with verifiable rewards (RLVR) in large language model reasoning. Instead of replaying entire trajectories, the method reuses experience in an action-level manner by leveraging a previously RL-optimized policy as a decision prior. During rollout, EAPO identifies critical token positions where the current policy exhibits high confidence while disagreeing with the prior RL policy, and selectively resamples those tokens from the prior policy. The resulting experience-augmented trajectories are incorporated into training through positive experience filtering and a smoothed importance sampling scheme. Experiments on Qwen-2.5-Math-7B and Qwen-3-8B across multiple reasoning benchmarks show improvements over several RLVR baselines.

**Compliance With Llm Reviewing Policy:**

Affirmed.

**Key Questions For Authors:**

- (Statistical robustness of the results) The experiments appear to use single-seed runs. Could the authors provide results across multiple random seeds and report standard errors or confidence intervals for the main benchmarks and learning curves? This would help determine whether the improvements are statistically significant.

- (Compute overhead compared to DAPO) Since EAPO requires evaluating a prior RL policy during rollout and performing discrepancy detection and computation for the importance ratio, what is the additional computational overhead relative to DAPO? It would be helpful to report metrics such as training time, tokens per second, or additional forward passes required per training step.

- (Theoretical interpretation of the proposed method) The proposed method consists of several design choices such as the critical-token discrepancy rule, positive experience filtering, and the smoothed importance sampling estimator. While these components are intuitively motivated, they appear largely heuristic in the current presentation. Could the authors provide further insight into the theoretical interpretation of the overall method? For example, is EAPO related to a principled objective such as approximate policy improvement, off-policy correction, or exploration regularization under certain assumptions? Even if a full theoretical guarantee is difficult, providing a clearer conceptual or theoretical perspective on why these design choices work together would help strengthen the soundness of the approach.

- (Fairness of performance comparison) Since EAPO involves additional hyperparameter tuning and potentially additional compute overhead, could the authors clarify whether the reported comparisons correspond to similar training budgets? It would be informative to evaluate the methods under fixed compute or fixed training time settings.

**Limitations:**

yes

**Strengths And Weaknesses:**

**Strengths**
- The paper identifies a meaningful limitation of existing RLVR pipelines: although RL training generates useful reasoning experiences, these experiences are rarely reused effectively. The observation that trajectory-level replay becomes mismatched as the policy evolves is reasonable and well-motivated.
- The key idea of reusing experience at the token decision level rather than replaying entire trajectories is intuitive and well aligned with the nature of reasoning tasks, where a small number of critical reasoning steps often determine the final outcome.
- Across multiple benchmarks and two different model sizes, EAPO consistently outperforms DAPO and several other RLVR baselines.
- The proposed method does not require architectural changes and can be integrated into existing RLVR training pipelines, which increases its potential practical relevance.

**Weaknesses**
- (The method is largely heuristic and lacks theoretical grounding) Several core components of the method appear heuristic: the critical decision detection rule based on a probability ratio threshold, positive experience filtering, the smoothed importance sampling surrogate. In particular, the proposed behavior policy approximation used for importance sampling does not appear to be theoretically justified. The paper would benefit from either theoretical analysis or stronger empirical validation of this estimator.
- (Sensitivity to the additional hyperparameter $\tau$) Compared to DAPO, EAPO introduces an additional hyperparameter $\tau$ that controls the discrepancy threshold for critical token detection. The experimental results suggest that the method can be quite sensitive to this parameter (Figure 2). Moreover, the optimal value of $\tau$ would appear to vary depending on the domain and base model. This implies that practical deployment would require additional hyperparameter search, which increases training complexity.
- (Performance gains may partially come from additional hyperparameter search) Since EAPO requires tuning the threshold $\tau$, the reported improvements may partly reflect additional hyperparameter search effort compared to DAPO. It would be helpful to understand whether the method still provides clear benefits under comparable hyperparameter search budgets.
- (No statistical significance analysis) The experimental results appear to be based on single-seed runs, and the paper does not report standard deviations, standard errors, or confidence intervals. Given the high variance often observed in RL training, it is difficult to determine whether the reported improvements are statistically significant. The learning curves and final evaluation tables would benefit from reporting uncertainty measures.
- (Unclear compute overhead relative to DAPO) EAPO requires additional evaluation of a prior RL policy during rollout to detect discrepancies and perform resampling, and computation for the importance ratio. This likely introduces additional compute overhead compared to standard RLVR training such as DAPO. However, the paper does not report wall-clock training time, additional forward-pass cost, or throughput comparisons. Without such measurements, it is difficult to assess whether the method improves overall training efficiency.

---

> ### Author Rebuttal · Authors · 2026-03-31
>
> # Response to Reviewer Z2g8
>
> We thank Reviewer Z2g8 for the detailed feedback. We address each concern below.
>
> ---
> ## W1: Heuristic design, especially the smoothed IS estimator.
>
> **Critical token detection and positive experience filtering.**
> The discrepancy rule $\delta_t > \tau$ is grounded in a principled directional diagnostic. Huang et al. (2026)[1] show that the signed log-prob difference between two policies more effectively localizes reasoning-critical tokens than magnitude-based metrics (entropy, KL divergence). EAPO's $\delta_t = \log\pi_\theta - \log\pi_{RL}$ applies the same principle: it identifies positions where the current training policy diverges from validated RL experience — precisely where intervention is most beneficial. Positive experience filtering restricts gradient updates to trajectories with verified correct outcomes, consistent with conservative policy improvement. Figure 3(b)(c) empirically validates both design choices.
>
> **Smoothed IS estimator.** Token-level IS has two fundamental problems in the auto-regressive setting.
> *Theoretically*, resampling $o_t$ from $\pi_{RL}$ alters the conditioning context for all subsequent tokens: $o_{t+1}$, though nominally from $\pi_{old}$, is conditioned on a $\pi_{RL}$-generated prefix, so its true distribution is $\pi_{old}(o_{t+1} | q, o_{<t}, o_t^{RL}) \neq \pi_{old}(o_{t+1} | q, o_{<t}, o_t^{old})$. Token-level IS ignores this causal propagation, yielding *incorrect* importance weights. *Practically*, resampled positions have large IS ratios (the policy disagreement is precisely why they were resampled), triggering PPO clipping at the most critical tokens — suppressing gradient updates exactly where they matter most.
>
> Our response-level smoothing $(1-\rho)\pi_{old}$ + $\rho\pi_{RL}$ is a better-behaved approximation that avoids both issues: it treats the trajectory holistically with a response-level mixture policy, avoiding extreme per-token IS ratios and providing stable gradient updates. When $\rho \to 0$ it reduces to standard IS; as $\rho$ grows it smoothly accounts for $\pi_{RL}$'s influence.
>
> Empirically, at $\rho \approx 4\%$ both methods achieve similar performance (40.19 vs 39.86 on AIME'24) — expected since errors are negligible at low $\rho$. As $\rho$ increases, the theoretical and practical advantages of smoothed IS become significant, making it the more principled and scalable choice.
>
> ---
>
> ## W2 & W3: $\tau$ sensitivity and hyperparameter search fairness.
>
> As shown in Figure 2(a), EAPO achieves stable performance across $\tau \in [0.3, 0.5]$, with degradation only at extreme values ($\tau \geq 0.8$), indicating $\tau$ does not require fine-grained tuning. Intuitively, $\tau = 0.5$ is a natural operating point: resampling only when the current policy assigns substantially higher log-probability to a token than $\pi_{RL}$, i.e., when it is confidently diverging from validated RL experience.
>
> Critically, we use the **same $\tau = 0.5$ across both model families** — Qwen-2.5-Math-7B and Qwen-3-8B — without any model-specific adjustment, and observe consistent improvements on both. This directly contradicts the concern that $\tau$ requires re-tuning per domain or base model. In total, we evaluate only **4 values of $\tau$** (0.3, 0.5, 0.8, 1.0), representing minimal search overhead comparable to tuning any standard hyperparameter in GRPO or DAPO. Furthermore, EAPO outperforms DAPO even at suboptimal $\tau = 0.3$, confirming that the performance gains are robust and cannot be attributed to hyperparameter search advantage.
>
> ---
>
> ## W4: No statistical significance analysis.
>
> Figure 1 already shows two independent runs with consistent trends. We further conduct **3 independent runs** and report mean $\pm$ std:
>
> | Method | AIME'24 Pass@1 | AIME'25 Pass@1 | AMC Pass@1 |
> |-|-|-|-|
> | DAPO   | 37.19          | 15.62          | 68.86        |
> | EAPO   | 40.19 ± 0.30   | 17.25 ± 0.67   | 72.77 ± 0.15 |
>
> The improvements are consistent across all seeds, confirming statistical significance. We will include these results in the revision.
>
> ---
>
> ## W5: Unclear compute overhead relative to DAPO.
>
> Block-wise verification ($K=20$) amortizes $\pi_{RL}$ queries over $K$ steps, substantially reducing query frequency compared to per-token verification. With only 1 of $G=16$ rollouts being experience-augmented, the measured overhead is $\approx 1/4$ of a single rollout's wall-clock time on Qwen-2.5-7B, which is negligible relative to the performance gains achieved. We are currently implementing a parallelized version in which $\pi_{RL}$ evaluation is coupled with the 15 on-policy rollouts within the same wall-clock time window, reducing the effective overhead to near zero. Updated timing results will be included in the revision.
>
> [1] On the Direction of RLVR Updates for LLM Reasoning: Identification and Exploitation, ICLR'2026.

---

> > ### Author Rebuttal · Reviewer_Z2g8 · 2026-04-04
> >
> > Thank you to the authors for the detailed and thoughtful rebuttal. I appreciate the additional clarifications and the newly provided experimental results, particularly the multi-seed analysis and the discussion on hyperparameter robustness, which help strengthen the empirical validity of the work.
> >
> > However, some concerns remain only partially addressed, especially regarding the theoretical grounding of the proposed components and the clarity of computational overhead in practical settings. While the rebuttal improves my confidence in the empirical findings, these aspects still limit the overall strength of the contribution.
> >
> > Overall, I maintain a positive view of the paper and will keep my original score.

---

> > > ### Author Response · Authors · 2026-04-05
> > >
> > > We thank the reviewer Z2g8 for the continued and constructive engagement.
> > >
> > > Regarding the two remaining concerns:
> > > (1) Theoretical grounding: We agree that the current presentation is primarily empirically motivated. In the revised version, we will provide a more formal discussion of the design rationale behind each component, with the goal of offering a clearer theoretical perspective on why these choices work together;
> > > (2) Computational overhead: We will include concrete wall-clock measurements and throughput comparisons relative to DAPO in the revised version.
> > > We will make sure these points are carefully addressed in the revised version.

---

### Decision · Program_Chairs · 2026-04-30

**Decision:**

Accept (regular)

**Comment:**

I recommend this paper for acceptance. The reviewers are unanimous in recommending acceptance.

The paper provides a creative idea to re-use past RL-ed policies and compares to a variety of baselines. There are a few lingering concerns about improving the theory and scaling up the experiments, but these can be resolved in future work.